# Design and Synthesis of Menthol and Thymol Derived Ciprofloxacin: Influence of Structural Modifications on the Antibacterial Activity and Anticancer Properties

**DOI:** 10.3390/ijms23126600

**Published:** 2022-06-13

**Authors:** Tomasz Szostek, Daniel Szulczyk, Jolanta Szymańska-Majchrzak, Michał Koliński, Sebastian Kmiecik, Dagmara Otto-Ślusarczyk, Aleksandra Zawodnik, Eliza Rajkowska, Kinga Chaniewicz, Marta Struga, Piotr Roszkowski

**Affiliations:** 1Biochemical Research Scientific Association, Chair and Department of Biochemistry, Medical University of Warsaw, 02-097 Warsaw, Poland; s076433@student.wum.edu.pl; 2Chair and Department of Biochemistry, Medical University of Warsaw, 02-097 Warsaw, Poland; jolanta.szymanska@wum.edu.pl (J.S.-M.); dotto@wum.edu.pl (D.O.-Ś.); mstruga@wum.edu.pl (M.S.); 3Bioinformatics Laboratory, Mossakowski Medical Research Institute, Polish Academy of Sciences, 5 Pawinskiego St., 02-106 Warsaw, Poland; mkolinski@imdik.pan.pl; 4Biological and Chemical Research Centre, Faculty of Chemistry, University of Warsaw, 02-089 Warsaw, Poland; sekmi@chem.uw.edu.pl; 5Department of Experimental and Clinical Pharmacology, Medical University of Warsaw, 02-091 Warsaw, Poland; aleksandra.zawodnik@gmail.com; 6Faculty of Chemistry, University of Warsaw, Pasteura 1, 02-093 Warsaw, Poland; e.rajkowska2@student.uw.edu.pl (E.R.); k.chaniewicz@student.uw.edu.pl (K.C.)

**Keywords:** antimicrobial, Ciprofloxacin, antibiotic, thymol, menthol

## Abstract

Sixteen new Ciprofloxacin derivatives were designed and successfully synthesized. In an in silico experiment, lipophilicity was established for obtained compounds. All compounds were screened for antimicrobial activity using standard and clinical strains. As for Gram-positive hospital microorganisms, all tested derivatives were active. Measured MICs were in the range 1–16 µg/mL, confirming high antimicrobial potency. Derivative **12** demonstrated activity against all standard Gram-positive *Staphylococci*, within the range of 0.8–1.6 µg/mL and was confirmed as the leading structure with MICs 1 µg/mL for *S. pasteuri* KR 4358 and *S. aureus* T 5591 (clinical strains). All compounds were screened for their in vitro cytotoxic properties via the MTT method. Three of the examined compounds (**3**, **11** and **16**) showed good activity against cancer cells, and in parallel were found not to be cytotoxic toward normal cells. Doxorubicin SI ranged 0.14–1.11 while the mentioned three ranged 1.9–3.4. Selected Ciprofloxacin derivatives were docked into the crystal structure of topoisomerase II (DNA gyrase) in complex with DNA (PDB ID: 5BTC). In summary, leading structures were established (**3**, **11**, **12** and **16**). We have observed poor results in preformed studies for disubstituted derivatives, suggesting that 3-oxo-4-carboxylic acid core is the active DNA-gyrase binding site, and when structural changes were made in this fragment, there was an observed decrease in antibacterial potency.

## 1. Introduction

Antibiotic resistance leads to various negative impacts on healthcare—both clinical and economic, mostly connected to delay or lack of effectiveness of antibiotic therapy. The economic impact is related to increased resource utilization, such as additional healthcare services, but also the costs of additional infection treatment and of treatment of health complications. The level of impact is correlated with the severity of the infection and virulence of strain, with the greatest impact of resistance to last line treatment options such as carbapenems and Vancomycin [1].

Based on the modelling of the global burden of antimicrobial resistance, the estimated number of deaths directly linked to infections with antimicrobial resistance in 2019 was 1.27 million, with the highest rate estimated for sub-Saharan Africa. According to the European Centre for Disease Prevention and Control (ECDC) estimations, the annual rate of infection caused by antimicrobial resistance in EU and European Economic Area (EEA) countries is 670,000 leading directly to 33,000 deaths. The above corresponded also with 874,541 DALYs (disability-adjusted life-years). It is also estimated that at the same time, by 2050 in the EU and EEA, there will be nearly 570 million additional hospital days per year. Most recent calculations for United States from 2019 show that fungi and bacteria resistant to antibiotics cause approximately 50,000 deaths [2,3,4,5]. All above estimations are directly linked with costs on three different levels: additional use of resources and healthcare costs, societal costs and costs of mortality and overall economic impact. Estimated values vary between identified publications, but each analysis shows a negative impact on above cost categories.

According to OECD data, calculated health care system costs for EU/EEA countries each year may be EUR 1.1 billion. Globally, the annual healthcare cost increase by 2050 may vary from USD 300 billion up to USD 1 trillion. Additional hospitalization costs are estimated for 10,000 to 40,000 USD per patient. For the most urgent threats assessed by CDC in United States, estimated healthcare costs in 2017 related to certain bacteria including ranged within: USD 281 million for Carbapenem-resistant *Acinetobacter*, USD 1 billion for *Clostridioides difficile* and USD 130 million for Carbapenem-resistant Enterobacteriaceae. Drug-resistant *Neisseria gonorrhoeae* leads to annual discounted lifetime USD 133.4 million in medical costs. Among serious threats, the bacterium MRSA generated USD 1.7 billion healthcare costs in US for 2017 [3,6,7]. Regarding the negative impact on the economy in general, OECD countries by 2050 may be exposed up to USD 2.9 trillion loss, which equals approximately 0.16% of their GDP. According to the World Bank report globally, annual GDP may be decreased by 1.1% even to 5% in the worst scenario by 2050. An identified systematic review of literature confirms the above data ranging up to USD 90 million healthcare costs per year and economic costs by 2050 up to $3 trillion global GDP loss [6,7,8].

Given this brief analysis of the economic impact of the phenomenon of antibiotic resistance, there is a great need to find new effective substances. Antibiotic resistance is a threat that has a large impact on public finances. The structural modification of the already existing antibiotics is one of the directions of the fight against antibiotic resistance. The formation of similar structures to reference antimicrobials can maintain the therapeutic effect at a similar level, while reducing the occurrence of resistance. Therefore, scientists all over the world are struggling to find new scaffolds that can effectively improve antimicrobial activity of lead structures and, in parallel, help to reduce bacterial resistance.

Essential oils are commonly used mixtures of naturally occurring substances to create a designed aroma or flavor. Some of their components may possess biological activities, therefore single compounds or groups are constantly studied. As popular phenolic essential oils, monoterpenes are well characterized and have been shown to present antimicrobial properties of their structural representatives such as: menthol, thymol, eugenol and carvacrol. From a number of published results, it is worth emphasizing confirmed activity of eugenol and thymol against bacterial strains including *Staphylococcus aureus* and *Pseudomonas aeruginosa* [9]. Activity of phenolic oils toward *Staphylococcus epidermidis*, *Escherichia coli*, *Pasteurella multocida*, *Neisseria gonorrhoeae*, methicillin-resistant *S. aureus*, and several other Gram-negative and Gram-positive bacteria was also confirmed [10]. Furthermore, some monoterpenes were found to be biofilm inhibitors, standalone or in combination, toward various bacteria species including *Cryptococcus*, *Salmonella*, *Staphylococci*, *Enterococcus*, *Escherichia*, *Porphyromonas*, and *Listeria* [11,12,13,14,15,16,17,18]. The mechanism of antimicrobial action of phenolic oils, including monoterpenes, was studied and established. In general, the ability of damaging bacterial biomembranes (toxic effect on structure of membranes) is pointed out as the main mode of action. However, it should be mentioned that several interactions with bacterial cells are responsible for antimicrobial activity. The mechanism of action should be linked with damage to membrane proteins, reduced ATP synthesis, increased membrane permeability and membrane fluidity causing leakage of ions, a decrease in the pH gradient across the cytoplasmic membrane [19,20]. Hydrophobic properties of phenolic oils are also an important aspect that is worth pointing out.

Menthol and thymol can be recognized as the most studied representatives of monoterpenes. The above mentioned properties were confirmed in dedicated studies [21,22]. Both structures may serve as moieties in modifications of the main chemical scaffold. One of the approaches in pursuit for new antimicrobial agents is the modification of marketed antibiotics. Ciprofloxacin’s structure was modified by various scaffolds to improve activity against bacterial strains, especially those resistant to commonly used antibiotics [23,24,25]. Parts of scientific reports are related to structural modifications of Ciprofloxacin by phenolic essential oils and specifically, menthol and thymol. Interesting results were presented by Dr. Mohamed’s group [26], who have established that a combination of essential oils and Ciprofloxacin is able to inhibit or eradicate biofilms in multidrug-resistant *Klebsiella pneumoniae*. Active components of thyme and peppermint essential oils were able to inhibit the biofilm of *K. pneumoniae*, alone or in combination with Ciprofloxacin. Most structural modifications of Ciprofloxacin are directed to N-4-piperazynyl framework and monoterpene structure is attached via linker (e.g., carboxymethyl) [27,28].

Encouraged by published results related to Ciprofloxacin-monoterpenes hybrids and our experience in research for new antimicrobial agents [29,30,31,32,33], we have decided to design and synthesize series of Ciprofloxacin derivatives with the utilization of menthol and thymol scaffolds. Furthermore, we wanted to explore the impact on antimicrobial activity, cytotoxicity and some of the physicochemical properties of novel compounds versus reference Ciprofloxacin.

## 2. Results and Discussion

### 2.1. In Silico Lipophilicity Calculation

It is commonly known that phenolic oils and their specific compounds, such as menthol and thymol, are rather hydrophobic. Since the designed compounds are Ciprofloxacin-based and menthol or thymol are used with suitable linker, it is worth checking how lipophilicity will vary from reference. Therefore, for all designed structures the n-octanol/water partition coefficient (log Po/w) was calculated, using iLOG descriptor [34] (Table 1).

It was found that n-octanol/water partition coefficient for all derivatives was much higher than reference, ranging from 3.81 to 7.72. However, it should be underlined that compounds **7**, **8**, **15** and **16** are structurally diverse from rest of the group (additional menthol or thymol substituent) and for them, lipophilicity is almost three times higher than Ciprofloxacin. If we compare results in suitable pairs (**1** and **9**, **2** and **10** etc.) there is no significant difference.

### 2.2. Chemistry

Synthesized structures were designed based on pharmacophore model [35]. Most models suggest that 3-oxo-4-carboxylic acid core is the active DNA-gyrase binding site and when structural changes were made in this fragment, there was an observed decrease in antibacterial potency [23,24,25]. However, modifications of the carboxylic group are still being studied. We believe that the attachment of fragments possessing antimicrobial properties related to different modes of action than quinolones may lead to antibiotics improvement or help to decrease the level of bacterial resistance.

We have decided to use menthol and thymol scaffolds to design and synthesize novel sixteen Ciprofloxacin derivatives. Twelve of the compounds should be recognized as adjustments linked to R7 of base quinolone structure, and four to R7 with additional carboxylic group modification (Figure 1) [35]. Menthol and thymol moieties were attached to Ciprofloxacin with the usage of diverse carboxylic linkers. Synthetic route is depicted below (Figure 1). Please see Appendix A for details related to synthetic pathway, methods and structure characterization of obtained compounds.

Adjustments to the reaction procedure helped to obtain three types of derivatives with the chain-increasing linker attached to piperazinyl part of Ciprofloxacin (R7 Figure 1), from acetyl to hexyl. Yields varied. Disubstituted compounds **7**, **8**, **15**, **16** were isolated next to suitable main compounds. For example, compound **6** was isolated as primary with 85% yield and as a result of simultaneous reaction with Ciprofloxacin carboxylic group, secondary product **8** was isolated (yield 8%). Details regarding synthetic procedures and spectral data can be found in point 4.2 and the Appendix A of the paper. All synthesized derivatives were transferred to biological activity evaluation.

### 2.3. Biological Studies

In Vitro Antibacterial Activity Studies

Obtained compounds were examined for antimicrobial potency. Firstly, all of them were screened for their minimal inhibitory concentrations (MICs) [36]. A suitable set of bacteria was used, including representative standard Gram-positive and Gram-negative rods (Table 2 and Table 3). For clarity, results are presented for menthol and thymol derivatives separately.

Disubstituted Ciprofloxacin derivatives **7**, **8**, **15**, **16** were not promising for further evaluation. This is the primary outcome that needs to be underlined. Thymol derivatives **15**, **16** showed slightly better results than two inactive menthol analogues, but level of MICs contrast with monosubstituted compounds and reference. Most probably, structural modification of 3-oxo-4-carboxylic acid core of Ciprofloxacin is responsible for antimicrobial activity decrease.

In general, the thymol group showed higher antimicrobial potency than menthol derivatives. Activity against both Gram-negative strains was weaker across all the investigated compounds. However, derivatives **3**, **4**, **11**, **12**, **13**, **14** were active against *E. coli* ATCC 25922, and observed MICs values ranged from 2 to 4 µg/mL. We found a very interesting observation regarding results linked to Gram-positive strains. All monosubstituted compounds exhibited a broad and high spectrum of activity. Excluding derivative **2**, obtained minimal inhibitory concentrations (range 0.8–8 µg/mL) should be considered very good. One of the compounds reached the antimicrobial potency level of the reference, Ciprofloxacin. Derivative **12** demonstrated activity against all standard Gram-positive *Staphylococci*, within the range of 0.8–1.6 µg/mL. All monosubstituted compounds were examined towards panel of clinical strains.

We have conducted experiment with the usage of four clinical Gram-positive and four clinical Gram-negative strains. Results (Table 4 and Table 5) are consistent comparing to those for standard bacteria strains. Results are presented in the same way as the above tables.

All examined compounds were rather inactive against strains of *E. coli* 510 and *P. aeruginosa* 659. Moderate activity was observed for derivatives **11**–**14**. Both menthol and thymol group showed similar potency towards *E. coli* 520 and 600, with MIC values within the range 4–64 µg/mL. Overall, used clinical Gram-negative strains were more susceptible to the studied compounds compared to standard rods. As for Gram-positive hospital microorganisms, all tested derivatives were active. Measured MICs were in the range 1–16 µg/mL, confirming high antimicrobial potency. Compound **12** was confirmed as the leading structure, with MICs 1 µg/mL for *S. pasteuri* KR 4358 and *S. aureus* T 5591.

What is interesting is that, in both evaluations, no trends were observed related to used linker (acetyl to hexyl). Best results were obtained for 1-cyclopropyl-6-fluoro-7-{4-[4-(2-isopropyl-5-methylphenoxy)-4-oxobutyl]piperazin-1-yl}-4-oxo-1,4-dihydroquinoline-3-carboxylic acid (**12**) from thymol group, containing butyl linker.

### 2.4. Molecular Docking

A set of the ligands studied in this work have been docked to the protein structure and also the Ciprofloxacin as the reference compound. Disubstituted derivatives were excluded from the experiment during the initial stage, because they were unable to fit to the binding pocket. In all ligand cases, the Ciprofloxacin scaffold is responsible for the ligand binding, and for many ligands the binding energy was on a similar level (see Table 6). Interestingly, calculated energies grow in groups (menthol **1**–**6** and thymol **9**–**14** derivatives), when the linker changes from oxoacetyl to oxohexyl. This trend will be explored in further studies.

The conformational entropy of the docking results measured by the size of the largest cluster clearly favors the structures of the reference compound Ciprofloxacin, ligands with propyl (ligands **3**, **11**) and then a butyl linker (ligands **4**, **12**). Selected ligands **4** and **12** are shown in Figure 2. In the case of ligand **12**, which showed the highest biological activity, carbonyl oxygen from the butyl linker creates a polar interaction with the edge of the binding pocket (see middle panel of Figure 2).

In summary, the docking results show the important role of the Ciprofloxacin scaffold for the ligands binding. In addition, the results suggest that thymol and menthol moieties may have a slightly destabilizing effect on the final complex, together with the linker (but not always), in comparison to Ciprofloxacin. Therefore, the role of thymol and menthol substituents may be different than enhancing the final interaction with the protein receptor. We assume that the mode of antimicrobial action may be dualistic. Introduction of the linker with attached menthol or thymol slightly affects pocket binding of the core scaffold (Ciprofloxacin). However, it increases lipophilicity (see Table 1) of the compounds and might produce additional interactions with bacterial biomembranes [19,20].

### 2.5. Anti-Cancer Activity

All compounds were screened for their in vitro cytotoxic properties via the MTT method. In this study panel, cancer cell and normal cell lines were used, specifically: human liver cancer (HepG2), human colon cancer (HCT-116), human primary colon cancer (SW480), human metastatic colon cancer (SW620), and human immortal keratinocyte cell line from adult human skin (HaCaT). Most of the tested derivatives exhibited moderate antiproliferative potency (Table 7). Yet, three of the examined compounds (**3**, **11** and **16**) showed good activity against cancer cells, and in parallel were found to be not cytotoxic toward normal cells. Selectivity indexes were higher in every case comparing to the reference. Doxorubicin SI ranged 0.14–1.11, while the mentioned compounds ranged 1.9–3.4. The level of reference activity was not reached by all three derivatives. What needs to be underlined is that Doxorubicin was highly cytotoxic toward all cancer cell lines, but for normal cells as well. The best selectivity index (3.4) was determined for compound **6**, with IC_50_ 29.5 ± 2.1 µM against human colon cancer cells (HCT-116) and with no cytotoxic effect on human immortal keratinocyte cell line from adult human skin (HaCaT).

It is worth to share other interesting findings. The highest cytotoxic effect was observed for **2**, with IC_50_ 16.5 ± 0.4 µM against human liver cancer cells (HepG2). Excluding derivative **6**, group of menthol derivatives were found to be cytotoxic for all cell lines, with most SI values < 1. In general, a similar outcome can be assigned to derivatives possessing oxoacetyl or oxypropyl linker. Disubstituted thymol derivatives were slightly cytotoxic against the normal cell line, while menthol analogues should be considered as cytotoxic. The derivative with the best antimicrobial activity, **12,** exhibited good result toward human immortal keratinocyte cell line from adult human skin (HaCaT). Compounds **3**, **11** and **16** showed the highest potency in the MTT assay and will be transferred for further testing.

## 3. Materials and Methods

### 3.1. Apparatus, Materials, and Analysis

Dichloromethane, methanol and dimethylformamide were supplied from Sigma Aldrich (Saint Louis, MO, USA). Ciprofloxacin (98%) was purchased from Acros Organics (Geel, Belgium), menthol (≥99%) and thymol (≥98.5%) were purchased from Sigma Aldrich (Saint Louis, MO, USA). All other chemicals were of analytical grade and were used without any further purification. The NMR spectra were recorded on a Bruker (Karlsruhe, Germany) AVANCE spectrometer (Bruker, Karlsruhe, Germany) operating at 300 MHz or 500 MHz for ^1^H NMR and at 75 MHz or 125 MHz for ^13^C NMR. The spectra were measured in CDCl_3_ or CDCl_3_\CD_3_OD, 9:1 mixture and are given as δ values (in ppm) relative to TMS. Mass spectral ESI measurements were carried out on LCT Micromass TOF HiRes apparatus (Micromass UK Limited, Manchester, UK). Melting points were determined on a Melting Point Meter KSP1D (A. Krüss Optronic, Hamburg, Germany) and were uncorrected. TLC analyses were performed on silica gel plates (Merck Kiesegel GF_254,_ Merck, Darmstadt, Germany) and visualized using UV light or iodine vapour. Column chromatography was carried out at atmospheric pressure using Silica Gel 60 (230–400 mesh, Merck, Darmstadt, Germany) and using dichloromethane/methanol (0–6%) mixture as eluent.

### 3.2. Ciprofloxacin Derivatives Preparation

#### 3.2.1. General Procedure for Synthesis of Ciprofloxacin Amides 1 and 9

To a magnetically stirred at room temperature suspension of Ciprofloxacin (0.318 g; 0.96 mmol, 1 eqv) in CH_2_Cl_2_ (50 mL), triethylamine (0.254 mL; 1.82 mmol, 1.9 eqv) was added and next, a solution of carboxylic acid chloride (1 eqv) in CH_2_Cl_2_ (2 mL) was dropped in over 2 min. After 1h of the reaction, a mixture of water (30 mL) and 3% HCl_aq_ solution were added to a pH equal to 3–4 and after separation of the phases, the water layer was extracted with CH_2_Cl_2_ (30 mL). The combined organic layers were washed with water (30 mL) and dried over Na_2_SO_4_. After evaporation of the solvent under reduced pressure, the product was isolated using column chromatography on silica gel and CH_2_Cl_2_:MeOH mixture (0–6% MeOH) as an eluent.

##### 1-cyclopropyl-6-fluoro-7-{4-[2-((1R,2S,5R)-2-isopropyl-5-methylcyclohexyloxy)acetyl]piperazin-1-yl}-4-oxo-1,4-dihydroquinoline-3-carboxylic acid (1)

White solid. Yield 92%. Mp = 185.7–189.6 °C.

^1^H NMR (CDCl_3_, 300 MHz) δ (ppm): 0.72 (d, *J* = 6.6 Hz, 3H), 0.89–0.93 (m, 9H), 1.15–1.34 (m, 6H), 1.53–1.61 (m, 2H), 2.06–2.17 (m, 2H), 3.01–3.18 (m, 1H), 3.27–3.30 (m, 4H), 3.51 (bs, 1H), 3.71–3.81 (m, 4H), 4.05–4.24 (m, 2H), 7.29 (d, *J* = 7.2 Hz, 1H), 7.85 (d, *J* = 12.9 Hz, 1H), 8.60 (s, 1H), 14.81 (s, 1H). ^13^C NMR (CDCl_3_, 75 MHz) δ (ppm): 8.2 (2xC), 16.1, 21.0, 22.3, 23.1, 25.5, 31.4, 34.3, 35.3, 40.0, 41.3, 45.2, 48.3, 49.3, 50.1, 68.3, 80.1, 105.1 (d, ^3^*J_C–F_* = 3.0 Hz), 107.9, 112.3 (d, ^2^*J_C–F_* = 23.3 Hz), 120.0 (d, ^3^*J_C–F_* = 7.5 Hz), 139.9, 145.4 (d, ^2^*J_C–F_* = 10.5 Hz), 147.4, 153.5 (d, ^1^*J_C–F_* = 249.8 Hz), 166.7, 168.6, 176.8 (d, ^4^*J_C–F_* = 2.3 Hz). HRMS (ESI) *m*/*z* 550.2693 calc. for C_29_H_38_FN_3_O_5_Na [M+Na]^+^; found 550.2681.

##### 1-cyclopropyl-6-fluoro-7-{4-[2-(2-isopropyl-5-methylphenoxy)acetyl]piperazin-1-yl}-4-oxo-1,4-dihydroquinoline-3-carboxylic acid (9)

White solid. Yield 81%. Mp = 232.8–234.0 °C.

^1^H NMR (CDCl_3_\CD_3_OD, 9:1 mixture, 300 MHz) δ (ppm): 1.18–1.20 (m, 2H), 1.21 (d, *J* = 7.0 Hz, 6H), 1.39–143 (m, 2H), 2.33 (s, 3H), 3.37–3.93 (m, 5H), 3.54–3.61 (m, 1H), 3.90 (t, *J* = 5.1 Hz, 4H), 4.76 (s, 2H), 6.72 (bs, 1H), 6.82 (d, *J* = 7.8 Hz, 1H), 7.14 (d, *J* = 7.5 Hz, 1H), 7.39 (d, *J* = 6.9 Hz, 1H), 8.04 (d, *J* = 12.9 Hz, 1H), 8.79 (s, 1H). ^13^C NMR (CDCl_3_\CD_3_OD, 9:1 mixture, 125 MHz) δ (ppm): 8.0 (2xC), 21.1, 22.8 (2xC), 26.2, 35.4, 41.7 (2xC), 45.2 (2xC), 67.9, 105.2 (d, ^3^*J_C–F_* = 3.0 Hz), 107.6, 112.2, 112.4 (d, ^2^*J_C–F_* = 23.3 Hz), 120.2 (d, ^3^*J_C–F_* = 7.5 Hz), 122.4, 126.1, 133.8, 136.6, 138.9, 145.2 (d, ^2^*J_C–F_* = 9.8 Hz), 147.7, 153.5 (d, ^1^*J_C–F_* = 249.8 Hz), 154.5, 167.3, 167.4, 177.0 (d, ^4^*J_C–F_* = 2.3 Hz). HRMS (ESI) *m*/*z* 544.2224 calc. for C_29_H_32_FN_3_O_5_Na [M+Na]^+^; found 544.2237.

#### 3.2.2. General Procedure for Synthesis of Menthol Derivatives of Ciprofloxacin

To a magnetically stirred at room temperature solution of appropriate menthol ester (1.2 mmol, 2 eqv) in DMF (10mL), Ciprofloxacin (0.60 mmol, 1 eqv) and NaHCO_3_ (1.2 mmol, 2 eqv) were added. The resulting suspension was stirred and heated at 70 °C (oil bath) for 20 h. The reaction mixture was evaporated under reduced pressure to dryness and to the residue, CH_2_Cl_2_ (20 mL) and water (80 mL) were added. Next, the 3% HCl_aq_ solution was added to a pH equal to 3–4 and after separation of the phases, the water layer was extracted with CH_2_Cl_2_ (20 mL). The combined organic layers were washed with water (10 mL) and dried over Na_2_SO_4_. After evaporation of the solvent under reduced pressure, the product was isolated using column chromatography on silica gel and CH_2_Cl_2_:MeOH mixture (0–4% MeOH) as an eluent.

##### 1-cyclopropyl-6-fluoro-7-{4-[2-((1R,2S,5R)-2-isopropyl-5-methylcyclohexyloxy)-2-oxoethyl]piperazin-1-yl}-4-oxo-1,4-dihydroquinoline-3-carboxylic acid (2)

White solid. Yield 63%. Mp = 214.5–215.4 °C.

^1^H NMR (CDCl_3_, 300 MHz) δ (ppm): 0.78 (d, *J* = 6.9 Hz, 3H), 0.84–0.96 (m, 1H), 0.90 (d, *J* = 2.4 Hz, 3H), 0.92 (d, *J* = 2.1 Hz, 3H), 0.96–1.14 (m, 2H), 1.18–1.23 (m, 2H), 1.37–1.55 (m, 4H), 1.69–1.72 (m, 2H), 1.82–1.89 (m, 1H), 1.99–2.04 (m, 1H), 2.84 (bs, 4H), 3.30(d, *J* = 2.7 Hz, 2H), 3.41 (t, *J* = 4.5 Hz, 4H), 3.51–3.61 (m, 1H), 4.77(dt, *J* = 4.5 Hz, 10.8 Hz, 1H), 7.37 (d, *J* = 7.2 Hz, 1H), 7.95 (d, *J* = 12.9 Hz, 1H), 8.72 (s, 1H), 15.00 (s, 1H). ^13^C NMR (CDCl_3_, 75 MHz) δ (ppm): 8.2 (2xC), 16.3, 20.7, 22.0, 23.4, 26.4, 31.4, 34.1, 35.3, 40.9, 46.9, 49.6 (d, ^4^*J_C–F_* = 4.5 Hz, 2xC), 52.5 (2xC), 59.3, 74.8, 104.9 (d, ^3^*J_C–F_* = 3.0 Hz), 108.0, 112.2 (d, ^2^*J_C–F_* = 23.3 Hz), 119.7 (d, ^3^*J_C–F_* = 7.5 Hz),139.0, 145.8 (d, ^2^*J_C–F_* = 10.5 Hz), 147.3,153.6 (d, ^1^*J_C–F_* = 247.5 Hz), 166.9, 169.6, 177.0 (d, ^4^*J_C–F_* = 2.3 Hz). HRMS (ESI) *m*/*z* 550.2693 calc. for C_29_H_38_FN_3_O_5_Na [M+Na]^+^; found 550.2685.

##### 1-cyclopropyl-6-fluoro-7-{4-[3-((1R,2S,5R)-2-isopropyl-5-methylcyclohexyloxy)-3-oxopropyl]piperazin-1-yl}-4-oxo-1,4-dihydroquinoline-3-carboxylic acid (3)

White solid. Yield 38%. Mp = 217.5–219.1 °C.

^1^H NMR (CDCl_3_, 500 MHz) δ (ppm): 0.78 (d, *J* = 7.0 Hz, 3H), 0.83–0.89 (m, 1H), 0.90 (d, *J* = 3.0 Hz, 3H), 0.91 (d, *J* = 2.5 Hz, 3H), 0.95–1.02 (m, 1H), 1.02–1.11 (m, 1H), 1.19–1.22 (m, 2H), 1.35–1.41 (m, 3H), 1.46–1.54 (m, 1H), 1.66–1.72 (m, 2H), 1.90–1.96 (m, 1H), 1.98–2.03 (m, 1H), 2.53 (t, *J* = 7.0 Hz, 2H), 2.70–2.72 (m, 4H), 2.79 (t, *J* = 7.0 Hz, 2H), 3.33(t, *J* = 5.0 Hz, 4H), 3.52–3.57 (m, 1H), 4.73(dt, *J* = 4.5 Hz, 11.0 Hz, 1H), 7.35 (d, *J* = 7.0 Hz, 1H), 7.98 (d, *J* = 13.0 Hz, 1H), 8.75 (s, 1H), 15.01 (s, 1H). ^13^C NMR (CDCl_3_, 125 MHz) δ (ppm): 8.2 (2xC), 16.4, 20.9, 22.1, 23.4, 26.2, 31.4, 32.9, 34.2, 35.3, 41.0, 47.0, 49.8 (d, ^4^*J_C–F_* = 4.5 Hz, 2xC), 52.5 (2xC), 53.6, 74.3, 104.7 (d, ^3^*J_C–F_* = 3.8 Hz), 108.1, 112.4 (d, ^2^*J_C–F_* = 22.5 Hz), 119.8 (d, ^3^*J_C–F_* = 8.8 Hz),139.1, 145.9 (d, ^2^*J_C–F_* = 10.0 Hz), 147.4,153.7 (d, ^1^*J_C–F_* = 250.0 Hz),167.0, 171.9, 177.1 (d, ^4^*J_C–F_* = 2.5 Hz). HRMS (ESI) *m*/*z* 564.2850 calc. for C_30_H_40_FN_3_O_5_Na [M+Na]^+^; found 564.22863.

##### 1-cyclopropyl-6-fluoro-7-{4-[4-((1R,2S,5R)-2-isopropyl-5-methylcyclohexyloxy)-4-oxobutyl]piperazin-1-yl}-4-oxo-1,4-dihydroquinoline-3-carboxylic acid (4)

White solid. Yield 70%. Mp = 269.4–270.8 °C.

^1^H NMR (CDCl_3_\CD_3_OD, 9:1 mixture, 500 MHz) δ (ppm):0.77 (d, *J* = 7.0 Hz, 3H), 0.86–0.89 (m, 1H), 0.91 (d, *J* = 5.0 Hz, 3H), 0.92 (d, *J* = 4.5 Hz, 3H), 0.97–1.02 (m, 1H), 1.03–1.10 (m, 1H), 1.22–1.25 (m, 2H), 1.38–1.42 (m, 1H), 1.46–1.48 (m, 2H), 1.50–1.54 (m, 1H), 1.68–1.72 (m, 2H), 1.80–1.86 (m, 1H), 1.95–1.99 (m, 1H), 2.22–2.28 (m, 2H), 2.51 (t, *J* = 6.0 Hz, 2H),3.24 (t, *J* = 5.5 Hz, 2H), 3.37 (bs, 4H), 3.65–3.69 (m, 1H), 3.82 (bs, 4H), 4.71(dt, *J* = 4.5 Hz, 11.0 Hz, 1H), 7.52 (d, *J* = 7.0 Hz, 1H), 7.91 (d, *J* = 12.5 Hz, 1H), 8.75 (s, 1H). ^13^C NMR (CDCl_3_\CD_3_OD, 9:1 mixture, 125 MHz) δ (ppm): 8.1 (2xC), 16.1, 20.5, 21.8, 23.2, 26.1, 26.2, 30.8, 31.3, 33.9, 35.6, 40.7, 46.5 (2xC), 46.7, 51.6 (2xC), 56.5, 75.1, 106.3 (d, ^3^*J_C–F_* = 2.5 Hz), 107.6, 112.1 (d, ^2^*J_C–F_* = 23.8 Hz), 120.8 (d, ^3^*J_C–F_* = 7.5 Hz), 138.8, 143.8 (d, ^2^*J_C–F_* = 10.0 Hz), 147.8,153.3 (d, ^1^*J_C–F_* = 248.8 Hz), 167.1, 171.7, 176.8 (d, ^4^*J_C–F_* = 2.5 Hz). HRMS (ESI) *m*/*z* 578.3006 calc. for C_31_H_42_FN_3_O_5_Na [M+Na]^+^; found 578.3021.

##### 1-cyclopropyl-6-fluoro-7-{4-[5-((1R,2S,5R)-2-isopropyl-5-methylcyclohexyloxy)-5-oxopentyl)piperazin-1-yl)-4-oxo-1,4-dihydroquinoline-3-carboxylic acid (5)

Pale beige solid. Yield 79%. Mp = 239.2–240.7 °C.

^1^H NMR (CDCl_3_, 500 MHz) δ (ppm): 0.77 (d, *J* = 7.0 Hz, 3H), 0.84–0.89 (m, 1H), 0.90 (d, *J* = 4.0 Hz, 3H), 0.92 (d, *J* = 3.5 Hz, 3H), 0.95–1.02 (m, 1H), 1.02–1.10 (m, 1H), 1.22–1.25 (m, 2H), 1.35–1.41 (m, 1H), 1.42–1.46 (m, 2H), 1.47–1.54 (m, 1H), 1.67–1.71 (m, 2H), 1.73–1.77 (m, 2H), 1.82–1.88 (m, 1H), 1.93–2.00 (m, 3H), 2.38 (t, *J* = 7.0 Hz, 2H), 3.01 (t, *J* = 8.5 Hz, 2H), 3.32 (bs, 4H), 3.57–3.61 (m, 1H), 3.77 (bs, 4H), 4.69 (dt, *J* = 4.0 Hz, 10.5 Hz, 1H), 7.37 (d, *J* = 7.0 Hz, 1H), 7.77 (d, *J* = 13.0 Hz, 1H), 8.65 (s, 1H), 14.85 (s, 1H). ^13^C NMR (CDCl_3_, 125 MHz) δ (ppm): 8.3 (2xC), 16.3, 20.7, 22.0, 22.3, 23.4, 23.7, 26.3, 31.4, 33.6, 34.1, 35.5, 40.9, 46.9, 47.0 (2xC), 51.8 (2xC), 57.4, 74.4, 105.6, 107.9, 112.0 (d, ^2^*J_C–F_* = 23.8 Hz), 120.1 (d, ^3^*J_C–F_* = 7.5 Hz), 138.8, 144.2 (d, ^2^*J_C–F_* = 10.0 Hz), 147.5, 153.2 (d, ^1^*J_C–F_* = 250.0 Hz), 166.5, 172.5, 176.6 (d, ^4^*J_C–F_* = 2.5 Hz). HRMS (ESI) *m*/*z* 592.3163 calc. for C_32_H_44_FN_3_O_5_Na [M+Na]^+^; found 592.3185.

##### 1-cyclopropyl-6-fluoro-7-{4-[6-((1R,2S,5R)-2-isopropyl-5-methylcyclohexyloxy)-6-oxohexyl]piperazin-1-yl}-4-oxo-1,4-dihydroquinoline-3-carboxylic acid (6)

Pale beige solid. Yield 85%. Mp = 210.8–212.4 °C.

^1^H NMR (CDCl_3_, 500 MHz) δ (ppm): 0.76 (d, *J* = 7.0 Hz, 3H), 0.83–0.89 (m, 1H), 0.90 (d, *J* = 3.5 Hz, 3H), 0.91 (d, *J* = 3.0 Hz, 3H), 0.94–1.01 (m, 1H), 1.02–1.10 (m, 1H), 1.22–1.25 (m, 2H), 1.35–1.41 (m, 1H), 1.42–1.46 (m, 4H), 1.47–1.53 (m, 1H), 1.66–1.72 (m, 4H), 1.80–1.87 (m, 1H), 1.87–1.98 (m, 3H), 2.33 (t, *J* = 7.0 Hz, 2H), 2.95 (t, *J* = 7.0 Hz, 2H), 3.28 (bs, 4H), 3.57–3.61 (m, 1H), 3.75 (bs, 4H), 4.68 (dt, *J* = 4.5 Hz, 11.0 Hz, 1H), 7.37 (d, *J* = 7.0 Hz, 1H), 7.78 (d, *J* = 13.0 Hz, 1H), 8.66 (s, 1H), 14.88 (s, 1H). ^13^C NMR (CDCl_3_, 125 MHz) δ (ppm): 8.3 (2xC), 16.3, 20.7, 22.0, 23.4, 24.1, 24.4, 26.3, 26.4, 31.4, 34.2, 34.3, 35.5, 40.9, 46.9, 47.2 (2xC), 51.9 (2xC), 57.7, 74.2, 105.6, 107.9, 111.9 (d, ^2^*J_C–F_* = 23.8 Hz), 120.1 (d, ^3^*J_C–F_* = 7.5 Hz), 138.8, 144.3 (d, ^2^*J_C–F_* = 11.3 Hz), 147.5, 153.2 (d, ^1^*J_C–F_* = 248.8 Hz), 166.6, 172.8, 176.6 (d, ^4^*J_C–F_* = 2.5 Hz). HRMS (ESI) *m*/*z* 606.3319 calc. for C_33_H_46_FN_3_O_5_Na [M+Na]^+^; found 606.3327.

##### 5-((1R,2S,5R)-2-isopropyl-5-methylcyclohexyloxy)-5-oxopentyl-1-cyclopropyl-6-fluoro-7-{4-[5-((1R,2S,5R)-2-isopropyl-5-methylcyclohexyloxy)-5-oxopentyl]piperazin-1-yl}-4-oxo-1,4-dihydroquinoline-3-carboxylate (7)

Solidifying oil. Yield 7%.

^1^H NMR (CDCl_3_, 300 MHz) δ (ppm): 0.75 (d, *J* = 4.8 Hz, 3H), 0.77 (d, *J* = 4.8 Hz, 3H), 0.84–0.92 (m, 14H), 0.95–1.16 (m, 6H), 1.28–1.51 (m, 6H), 1.64–1.72 (m, 8H), 1.78–1.90 (m, 6H), 1.94–2.02 (m, 2H), 2.31–2.39 (m, 4H), 2.45 (t, *J* = 4.2 Hz, 2H), 2.65 (t, *J* = 4.8 Hz, 4H), 3.29 (t, *J* = 4.8 Hz, 4H), 3.38–3.46 (m, 1H), 4.32 (t, *J* = 6.0 Hz, 2H), 4.63–4.74 (m, 2H), 7.27 (d, *J* = 7.2 Hz, 1H), 8.04 (d, *J* = 13.5 Hz, 1H), 8.52 (s, 1H). ^13^C NMR (CDCl_3_, 75 MHz) δ (ppm): 8.1 (2xC), 16.3, 16.3, 20.7, 20.8, 21.7, 22.0, 23.0, 23.4, 23.4, 26.2, 26.2, 26.3, 28.2, 31.4, 34.2, 34.2, 34.5, 34.5, 40.9, 41.0, 47.0 (d, ^4^*J_C–F_* = 3.0 Hz), 49.9 (d, ^4^*J_C–F_* = 3.8 Hz), 52.9, 58.0, 64.3, 74.0, 104.7 (d, ^4^*J_C–F_* = 2.3 Hz), 110.3, 113.3 (d, ^2^*J_C–F_* = 23.3 Hz), 122.9 (d, ^3^*J_C–F_* = 7.5 Hz), 138.0, 144.6 (d, ^2^*J_C–F_* = 10.5 Hz), 148.0, 153.4 (d, ^1^*J_C–F_* = 247.5 Hz), 165.7, 173.0, 173.0 (d, ^4^*J_C–F_* = 1.5 Hz), 173.1. HRMS (ESI) *m*/*z* 830.5095 calc. for C_47_H_70_FN_3_O_7_Na [M+Na]^+^; found 830.5072.

##### 6-((1R,2S,5R)-2-isopropyl-5-methylcyclohexyloxy)-6-oxohexyl-1-cyclopropyl-6-fluoro-7-{4-[6-((1R,2S,5R)-2-isopropyl-5-methylcyclohexyloxy)-6-oxohexyl]piperazin-1-yl}-4-oxo-1,4-dihydroquinoline-3-carboxylate (8)

Solidifying oil. Yield 8%.

^1^H NMR (CDCl_3_, 300 MHz) δ (ppm): 0.74 (d, *J* = 4.8 Hz, 3H), 0.77 (d, *J* = 4.8 Hz, 3H), 0.84–0.94 (m, 14H), 0.97–1.19 (m, 6H), 1.30–1.43 (m, 6H), 1.46–1.59 (m, 6H), 1.62–1.72 (m, 8H), 1.79–1.89 (m, 4H), 1.94–2.02 (m, 2H), 2.28–2.34 (m, 4H), 2.43 (t, *J* = 4.5 Hz, 2H), 2.67 (t, *J* = 4.2 Hz, 4H), 3.29 (t, *J* = 4.5 Hz, 4H), 3.40–3.46 (m, 1H), 4.31 (t, *J* = 6.6 Hz, 2H), 4.62–4.73 (m, 2H), 7.26 (d, *J* = 6.3 Hz, 1H), 8.03 (d, *J* = 13.2 Hz, 1H), 8.51 (s, 1H). ^13^C NMR (CDCl_3_, 75 MHz) δ (ppm): 8.1 (2xC), 16.3, 16.3, 20.7, 20.8, 22.0, 22.0, 23.4, 24.8, 25.0, 25.7, 26.2, 26.3, 26.5, 27.0, 28.4, 31.3, 34.2, 34.4, 34.5, 34.6, 40.9, 40.9, 47.0 (d, ^4^*J_C–F_* = 2.3 Hz), 49.9 (d, ^4^*J_C–F_* = 4.5 Hz), 52.9, 58.3, 64.6, 73.9, 73.9, 104.7 (d, ^4^*J_C–F_* = 3.0 Hz), 110.3, 113.2 (d, ^2^*J_C–F_* = 23.3 Hz), 122.9 (d, ^3^*J_C–F_* = 6.8 Hz), 138.0, 144.5 (d, ^2^*J_C–F_* = 10.5 Hz), 148.0, 153.4 (d, ^1^*J_C–F_* = 247.5 Hz), 165.8, 173.0 (d, ^4^*J_C–F_* = 1.5 Hz), 173.1, 173.2. HRMS (ESI) *m*/*z* 858.5408 calc. for C_49_H_74_FN_3_O_7_Na [M+Na]^+^; found 858.5427.

#### 3.2.3. General Procedure for Synthesis of Thymol Derivatives of Ciprofloxacin

To a magnetically stirred at room temperature solution of appropriate menthol ester (1.2 mmol, 2 eqv) in DMF (10 mL) Ciprofloxacin (0.60 mmol, 1 eqv) and NaHCO_3_ (1.2 mmol, 2 eqv) were added. The resulting suspension was stirred and heated at 70 °C (oil bath) for 24 h. The reaction mixture was evaporated under reduced pressure to dryness and to residue CH_2_Cl_2_ (20 mL) and water (80 mL) were added. Next, the 3% HCl_aq_ solution was added to a pH equal to 3–4 and after separation of the phases, the water layer was extracted with CH_2_Cl_2_ (20 mL). The combined organic layers were washed with water (10 mL) and dried over Na_2_SO_4_. After evaporation of the solvent under reduced pressure, the product was isolated using column chromatography on silica gel and CH_2_Cl_2_:MeOH mixture (0–6% MeOH) as an eluent.

##### 1-cyclopropyl-6-fluoro-7-{4-[2-(2-isopropyl-5-methylphenoxy)acetyl]piperazin-1-yl}-4-oxo-1,4-dihydroquinoline-3-carboxylic acid (10)

Pale beige solid. Yield 65%. Mp = 240.2–241.8 °C.

^1^H NMR (CDCl_3_\CD_3_OD, 9:1 mixture, 500 MHz) δ (ppm): 1.18–1.20 (m, 2H), 1.21 (d, *J* = 7.0 Hz, 6H), 1.39–143 (m, 2H), 2.33 (s, 3H), 2.95 (t, *J* = 4.0 Hz, 4H), 3.38–3.40 (m, 1H), 3.45 (t, *J* = 4.5 Hz, 4H), 3.56–3.60 (m, 1H), 3.63 (s, 2H), 6.83 (bs, 1H), 7.04–7.06 (m, 1H), 7.22 (d, *J* = 8.0 Hz, 1H), 7.40 (d, *J* = 7.0 Hz, 1H), 7.98 (d, *J* = 13.0 Hz, 1H), 8.76 (s, 1H). ^13^C NMR (CDCl_3_\CD_3_OD, 9:1 mixture, 125 MHz) δ (ppm): 8.1 (2xC), 20.7, 22.9 (2xC), 27.1, 35.4, 49.5 (d, ^4^*J_C–F_* = 2.0 Hz, 2xC), 52.4 (2xC), 58.8, 105.0 (d, ^3^*J_C–F_* = 3.8 Hz), 107.7, 112.2 (d, ^2^*J_C–F_* = 23.8 Hz), 119.8 (d, ^3^*J_C–F_* = 7.5 Hz), 122.4, 126.5, 127.3, 136.6, 136.8, 139.1, 145.7 (d, ^2^*J_C–F_* = 10.0 Hz), 147.3, 147.5, 153.6 (d, ^1^*J_C–F_* = 250.0 Hz), 167.3, 168.9, 177.0 (d, ^4^*J_C–F_* = 2.5 Hz). HRMS (ESI) *m*/*z* 544.2224 calc. for C_29_H_32_FN_3_O_5_Na [M+Na]^+^; found 544.2240.

##### 1-cyclopropyl-6-fluoro-7-{4-[3-(2-isopropyl-5-methylphenoxy)-3-oxopropyl]piperazin-1-yl}-4-oxo-1,4-dihydroquinoline-3-carboxylic acid (11)

White solid. Yield 45%. Mp = 171.6–173.0 °C.

^1^H NMR (CDCl_3_, 300 MHz) δ (ppm): 1.17–1.23 (m, 2H), 1.20 (d, *J* = 6.9 Hz, 6H), 1.36–143 (m, 2H), 2.31 (s, 3H), 2.77–2.84 (m, 6H), 2.94 (t, *J* = 6.0 Hz, 2H), 3.03–3.13 (m, 1H), 3.38 (t, *J* = 4.8 Hz, 4H), 3.52–3.59 (m, 1H), 6.83 (bs, 1H), 7.01–7.05 (m, 1H), 7.21 (d, *J* = 7.8 Hz, 1H), 7.36 (d, *J* = 7.2 Hz, 1H), 7.95 (d, *J* = 13.2 Hz, 1H), 8.72 (s, 1H), 15.00 (s, 1H). ^13^C NMR (CDCl_3_, 75 MHz) δ (ppm): 8.2 (2xC), 20.8, 23.2 (2xC), 26.8, 32.7, 35.3, 49.8 (d, ^4^*J_C–F_* = 5.3 Hz, 2xC), 52.6 (2xC), 67.1, 104.8 (d, ^3^*J_C–F_* = 3.8 Hz), 108.0, 112.3 (d, ^2^*J_C–F_* = 23.3 Hz), 119.7 (d, ^3^*J_C–F_* = 8.3 Hz), 122.6, 126.4, 127.2, 136.6, 137.0, 139.0, 145.8 (d, ^2^*J_C–F_* = 9.8 Hz), 147.3, 147.8, 153.6 (d, ^1^*J_C–F_* = 249.8 Hz), 166.9, 171.0, 177.0 (d, ^4^*J_C–F_* = 2.3 Hz). HRMS (ESI) *m*/*z* 558.2380 calc. for C_30_H_34_FN_3_O_5_Na [M+Na]^+^; found 558.22364.

##### 1-cyclopropyl-6-fluoro-7-{4-[4-(2-isopropyl-5-methylphenoxy)-4-oxobutyl]piperazin-1-yl}-4-oxo-1,4-dihydroquinoline-3-carboxylic acid (12)

White solid. Yield 59%. Mp = 244.1–245.7 °C.

^1^H NMR (CDCl_3_\CD_3_OD, 9:1 mixture, 500 MHz) δ (ppm): 1.20 (d, *J* = 6.5 Hz, 6H), 1.22–1.24 (m, 2H), 1.42–1.43 (m, 2H), 2.29–2.32 (m, 5H), 2.82 (t, *J* = 6.5 Hz, 2H), 2.91–2.96 (m, 1H), 3.12–3.21 (m, 2H), 3.29–3.44 (m, 4H), 3.59–3.62 (m, 1H), 3.73 (bs, 4H), 6.82 (s, 1H), 7.05 (d, *J* = 8.0 Hz, 1H), 7.21 (d, *J* = 8.0 Hz, 1H), 7.45 (d, *J* = 7.5 Hz, 1H), 7.90 (d, *J* = 12.5 Hz, 1H), 8.73 (s, 1H). ^13^C NMR (CDCl_3_\CD_3_OD, 9:1 mixture, 125 MHz) δ (ppm): 8.2 (2xC), 19.5, 20.7, 23.0 (2xC), 27.1, 31.0, 35.5, 47.2 (2xC), 51.9 (2xC), 56.6, 105.9, 107.7, 112.2 (d, ^2^*J_C–F_* = 23.8 Hz), 120.6 (d, ^3^*J_C–F_* = 7.5 Hz), 122.4, 126.5, 127.3, 136.6, 136.7, 138.2, 144.3 (d, ^2^*J_C–F_* = 11.3 Hz), 147.5, 147.7, 153.4 (d, ^1^*J_C–F_* = 250.0 Hz), 167.0, 171.4, 176.8 (d, ^4^*J_C–F_* = 2.5 Hz). HRMS (ESI) *m*/*z* 550.2717 calc. for C_31_H_37_FN_3_O_5_ [M+H]^+^; found 550.2736.

##### 1-cyclopropyl-6-fluoro-7-{4-[5-(2-isopropyl-5-methylphenoxy)-5-oxopentyl]piperazin-1-yl}-4-oxo-1,4-dihydroquinoline-3-carboxylic acid (13)

White solid. Yield 81%. Mp = 212.3–213.9 °C.

^1^H NMR (CDCl_3_, 500 MHz) δ (ppm): 1.19 (d, *J* = 6.5 Hz, 6H), 1.21–1.23 (m, 2H), 1.40–1.46 (m, 2H), 1.83–1.95 (m, 4H), 2.11–2.19 (m, 2H), 2.32 (s, 3H), 2.68–2.73 (m, 2H), 2.90–2.96 (m, 1H), 3.23 (bs, 4H), 3.59–3.65 (m, 1H), 3.88 (bs, 4H), 6.81 (s, 1H), 7.03 (d, *J* = 7.5 Hz, 1H), 7.20 (d, *J* = 8.0 Hz, 1H), 7.38 (bs, 1H), 7.71 (d, *J* = 12.5 Hz, 1H), 8.61 (s, 1H), 14.85 (s, 1H). ^13^C NMR (CDCl_3_, 125 MHz) δ (ppm): 8.3 (2xC), 20.8, 21.9, 23.0 (2xC), 27.0, 27.1, 33.1, 35.6, 46.3 (d, ^4^*J_C–F_* = 2.5 Hz, 2xC), 51.5 (2xC), 57.1, 105.9, 107.7, 111.9 (d, ^2^*J_C–F_* = 25.0 Hz), 120.2 (d, ^3^*J_C–F_* = 7.5 Hz), 122.6, 126.4, 127.2, 136.6, 136.8, 138.7, 143.6 (d, ^2^*J_C–F_* = 10.0 Hz), 147.5, 147.6, 153.1 (d, ^1^*J_C–F_* = 248.8 Hz), 166.4, 171.7, 176.5 (d, ^4^*J_C–F_* = 2.5 Hz). HRMS (ESI) *m*/*z* 586.2693 calc. for C_32_H_38_FN_3_O_5_Na [M+Na]^+^; found 586.2679.

##### 1-cyclopropyl-6-fluoro-7-{4-[6-(2-isopropyl-5-methylphenoxy)-6-oxohexyl]piperazin-1-yl}-4-oxo-1,4-dihydroquinoline-3-carboxylic acid (14)

White solid. Yield 64%. Mp = 185.4–187.1 °C.

^1^H NMR (CDCl_3_, 500 MHz) δ (ppm): 1.19 (d, *J* = 6.5 Hz, 6H), 1.21–1.23 (m, 2H), 1.40–1.44 (m, 2H), 1.51–1.58 (m, 2H), 1.81–1.87 (m, 2H), 1.88–1.93 (m, 2H), 2.31 (s, 3H), 2.63 (t, *J* = 7.5 Hz, 2H), 2.88 (bs, 2H), 2.92–2.98 (m, 1H), 3.17 (bs, 4H), 3.54–3.59 (m, 1H), 3.67 (bs, 4H), 6.80 (s, 1H), 7.02 (d, *J* = 8.0 Hz, 1H), 7.20 (d, *J* = 8.0 Hz, 1H), 7.36 (d, *J* = 7.0 Hz, 1H), 7.82 (d, *J* = 12.5 Hz, 1H), 8.67 (s, 1H), 14.85 (s, 1H). ^13^C NMR (CDCl_3_, 125 MHz) δ (ppm): 8.3 (2xC), 20.8, 23.0 (2xC), 24.4, 24.6, 26.5, 27.1, 33.9, 35.4, 47.7 (2xC), 52.1 (2xC), 57.7, 105.4 (d, ^3^*J_C–F_* = 2.5 Hz), 107.9, 112.1 (d, ^2^*J_C–F_* = 23.8 Hz), 120.0 (d, ^3^*J_C–F_* = 7.5 Hz), 122.6, 126.4, 127.1, 136.5, 136.9, 138.9, 144.6 (d, ^2^*J_C–F_* = 10.0 Hz), 147.5, 147.7, 153.3 (d, ^1^*J_C–F_* = 250.0 Hz), 166.6, 172.1, 176.7 (d, ^4^*J_C–F_* = 2.5 Hz). HRMS (ESI) *m*/*z* 578.3030 calc. for C_33_H_41_FN_3_O_5_ [M+H]^+^; found 578.3049.

##### 2-(2-isopropyl-5-methylphenoxy)-2-oxobutyl-1-cyclopropyl-6-fluoro-7-{4-[2-(2-isopropyl-5-methylphenoxy)-2-oxobutyl]piperazin-1-yl}-4-oxo-1,4-dihydroquinoline-3-carboxylate (15)

White solid. Yield 21%. Mp = 105.5–107.2 °C.

^1^H NMR (CDCl_3_, 500 MHz) δ (ppm): 1.06–1.09 (m, 2H), 1.16 (d, *J* = 7.0 Hz, 6H), 1.20 (d, *J* = 7.0 Hz, 6H), 1.24–1.28 (m, 2H), 1.98–2.04 (m, 2H), 2.23 (s, 3H), 2.24–2.28 (m, 2H), 2.31 (s, 3H), 2.56 (t, *J* = 7.0 Hz, 2H), 2.68 (t, *J* = 7.0 Hz, 2H), 2.71 (t, *J* = 4.0 Hz, 4H), 2.83 (t, *J* = 7.5 Hz, 2H), 2.93–3.01 (m, 2H), 3.30 (t, *J* = 4.5 Hz, 4H), 3.33–3.38 (m, 1H), 4.43 (t, *J* = 6.5 Hz, 2H), 6.78 (s, 1H), 6.82 (s, 1H), 6.99 (d, *J* = 8.0 Hz, 1H), 7.02 (d, *J* = 7.5 Hz, 1H), 7.17 (d, *J* = 7.5 Hz, 1H), 7.20 (d, *J* = 8.0 Hz, 1H), 7.26 (d, *J* = 7.5 Hz, 1H), 8.02 (d, *J* = 13.5 Hz, 1H), 8.48 (s, 1H). ^13^C NMR (CDCl_3_, 125 MHz) δ (ppm): 8.0 (2xC), 20.7, 20.8, 22.0, 23.0 (2xC), 23.0 (2xC), 24.3, 27.0, 27.1, 31.2, 32.0, 34.5, 49.9 (d, ^4^*J_C–F_* = 5.0 Hz, 2xC), 52.9 (2xC), 57.4, 63.7, 104.7 (d, ^3^*J_C–F_* = 2.5 Hz), 110.0, 113.2 (d, ^2^*J_C–F_* = 22.5 Hz), 122.6, 122.7, 122.9 (d, ^3^*J_C–F_* = 7.5 Hz), 126.3, 126.4, 127.0, 127.1, 136.4, 136.5, 136.9, 137.0, 138.0, 144.5 (d, ^2^*J_C–F_* = 10.0 Hz), 147.8 (2xC), 148.2, 153.4 (d, ^1^*J_C–F_* = 246.3 Hz), 165.6, 171.9, 172.2, 173.1 (d, ^4^*J_C–F_* = 2.5 Hz). HRMS (ESI) *m*/*z* 768.4024 calc. for C_45_H_55_FN_3_O_7_ [M+H]^+^; found 768.4011.

##### 4-(2-isopropyl-5-methylphenoxy)-4-oxohexyl-1-cyclopropyl-6-fluoro-7-{4-[4-(2-isopropyl-5-methylphenoxy)-4-oxohexyl]piperazin-1-yl}-4-oxo-1,4-dihydroquinoline-3-carboxylate (16)

Solidifying oil. Yield 14%.

^1^H NMR (CDCl_3_, 500 MHz) δ (ppm): 1.09–1.12 (m, 2H), 1.18 (d, *J* = 7.0 Hz, 6H), 1.19 (d, *J* = 7.0 Hz, 6H), 1.26–1.30 (m, 2H), 1.47–1.53 (m, 2H), 1.58–1.66 (m, 4H), 1.80–1.90 (m, 6H), 2.29 (s, 3H), 2.31 (s, 3H), 2.47 (t, *J* = 7.5 Hz, 2H), 2.59–2.63 (m, 4H), 2.68 (t, *J* = 5.0 Hz, 4H), 2.92–2.99 (m, 2H), 3.30 (t, *J* = 5.0 Hz, 4H), 3.36–3.41 (m, 1H), 4.35 (t, *J* = 6.5 Hz, 2H), 6.79 (s, 1H), 6.80 (s, 1H), 6.99–7.03 (m, 2H), 7.17–7.20 (m, 2H), 7.26 (d, *J* = 7.0 Hz, 1H), 7.26 (d, *J* = 7.5 Hz, 1H), 8.02 (d, *J* = 13.0 Hz, 1H), 8.50 (s, 1H). ^13^C NMR (CDCl_3_, 125 MHz) δ (ppm): 8.1 (2xC), 20.8, 20.8, 23.0 (2xC), 23.0 (2xC), 24.7, 24.9, 25.6, 26.5, 27.0, 27.0, 27.1, 28.5, 34.2, 34.2, 34.4, 49.9 (d, ^4^*J_C–F_* = 5.0 Hz, 2xC), 53.0 (2xC), 58.3, 64.5, 104.7 (d, ^3^*J_C–F_* = 2.5 Hz), 110.3, 113.2 (d, ^2^*J_C–F_* = 23.8 Hz), 122.7, 122.7, 122.9 (d, ^3^*J_C–F_* = 7.5 Hz), 126.3, 126.4, 127.0, 127.0, 136.4, 136.5, 136.9, 137.0, 138.0, 144.5 (d, ^2^*J_C–F_* = 11.3 Hz), 147.8, 147.9, 148.1, 153.4 (d, ^1^*J_C–F_* = 247.5 Hz), 165.9, 172.3, 172.3, 173.1 (d, ^4^*J_C–F_* = 2.5 Hz). HRMS (ESI) *m*/*z* 846.4469 calc. for C_49_H_62_FN_3_O_7_Na [M+Na]^+^; found 846.4437.

### 3.3. Biological Assays

The antimicrobial assays were conducted using reference strains of bacteria derived from international microbe collections: American Type Culture Collection (ATCC) and National Collection of Type Culture (NCTC). The following standard strains of bacteria were used: Gram-positive—*Staphylococcus aureus* NCTC 4163, *Staphylococcus aureus* ATCC 25923, *Staphylococcus aureus* ATCC 6538, *Staphylococcus aureus* ATCC 29213, *Staphylococcus epidermidis* ATCC 12228, *Staphylococcus epidermidis* ATCC 35984, Gram-negative: *Escherichia coli* ATCC 25922, *Pseudomonas aeruginosa* ATCC 15442. The clinical strains of bacteria used in this study were: Gram-positive: *Staphylococcus epidermidis* KR 4243, *Staphylococcus pasteuri KR* 4358, *Staphylococcus aureus* T 5595, *Staphylococcus aureus* T 5591 and Gram-negative: *Escherichia coli* 520, *Escherichia coli* 600, *Escherichia coli* 510 and *Pseudomonas aeruginosa* 659 were obtained from the collection of the Department of Pharmaceutical Microbiology, Medical University of Warsaw, Poland and they were isolated from different biological materials taken from patients hospitalized in Warsaw Medical University hospitals. Antimicrobial activity was examined by the Minimal Inhibitory Concentration (MIC) method under standard procedures provided by CLSI with some modifications. MIC was determined by the two-fold serial broth microdilution method in 96-well microtitration plates using Mueller–Hinton II broth medium (Becton Dickinson, Franklin Lakes, NJ, USA). The final inoculum of all studied bacteria was 10^6^ CFU/mL (colony forming unit per millilitre). The stock solution of tested compounds was prepared in dimethyl sulfoxide (DMSO) and diluted to a maximum of 1% of solvent content with a sterile medium. The MIC value recorded is defined as the lowest concentration of the tested antimicrobial agents (expressed in µg/mL) that inhibit the visible growth of the microorganism after 19 h of incubation at 35 °C.

Descriptions related to the conducted biological studies including cell culture, suitable conditions, and methodology were presented in our previous paper [29].

### 3.4. Molecular Docking Studies

A set of 13 ligands (Table 6), including Ciprofloxacin and its derivatives, was docked into the crystal structure of topoisomerase II (DNA gyrase) in complex with DNA (PDB ID: 5BTC [38].

Disubstituted ligands were excluded from the docking because the preliminary analysis showed that substituents at the carboxyl residue (of the Ciprofloxacin scaffold) prevent the binding of disubstituted ligands to a tight pocket. The analysis of the structure of the experimental Ciprofloxacin complex shows that the free carboxyl residue, closely fitting the binding gap, forms important stabilizing interactions with the protein residues. Importantly, these observations are consistent with poor binding of disubstituted ligands presented in the experimental results; see Table 6.

Ligand structures were generated using the Automated Topology Builder (ATB version 2.2) server [39]. Molecular docking and data analysis were performed using AutoDock4 (v. 4.2) and AutoDockTools [40]. For each receptor-ligand complex, the docking procedure included 1000 independent docking simulations performed using a genetic algorithm with local search (GA-LS), resulting in 1000 conformers with the lowest binding energy. Structural clustering (with RMSD cutoff at 3 Å) was then applied to identify the most preferred ligand binding modes. The central structure of the largest cluster was selected as the final ligand docked structure for each complex.

### 3.5. Anti-Cancer Studies

#### 3.5.1. Cell Line and Culture

The human cell lines SW480 (primary colon cancer), SW620 (lymph node metastatic colon cancer from the same patient as primary cancer cells), HCT116 (colon carcinoma), HepG2 (liver cancer), and HaCaT (immortalized keratinocytes) were obtained from the American Type Culture Collection (ATCC, Rockville, MD, USA). The SW480, SW620 and HCT116 cells were grown in MEM (ThermoSci, Waltham, MA, USA), HepG2 and HaCaT in DMEM High Glucose (Biowest SAS, Nuaillé, France) supplemented with 10% foetal bovine serum (FBS), HEPES (20 mM), and antibiotics (100 U mL^−1^ of Penicillin and 100 μg mL^−1^ of Streptomycin). The cells were incubated in a humidified incubator at 37 °C/5% CO_2_, until 80–90% confluence was reached.

#### 3.5.2. MTT Assay

The cell viability was assessed by using of MTT salt [3-(4,5-dimethylthiazol-2-yl)-2,5-diphenyltetrazolium bromide] converted by mitochondrial dehydrogenase, occurring in living cells. The cells were seeded in 96-well plates at a density of 1 × 10^4^ cells per well and allowed to adhere for 24 h at 37 °C in a CO_2_ humidified incubator. Then, the medium was removed and a fresh medium with various concentrations of tested compounds (from 10 µM to120 µM) was added. The untreated cells were used as the control.

After 72 h incubation, the medium was replaced with 200 µL per well of free-serum medium containing 0.5 mg mL^−1^ MTT and incubated for 4 h at 37 °C in a CO_2_ humidified incubator. Subsequently, the medium was removed and dimethyl sulfoxide (DMSO) with isopropanol (1:1) was added to dissolve the formazan crystals. The optical density was measured using UVM 340 reader (ASYS Hitech GmbH, Eugendorf, Austria) at a wavelength of 570 nm. The experiments were repeated three times. The cell viability was calculated as the percent of MTT reduced in treated cells versus control cells (untreated cells). The number of viable cells cultured without tested compounds was assumed as 100%. A decreased relative MTT level indicates decreased cell viability. The IC_50_ values were estimated using CompuSyn version 1.0.

## 4. Conclusions

Structural modifications of Ciprofloxacin were designed and synthesized. Sixteen new derivatives were screened for antimicrobial activity. Furthermore, an MTT assay was performed to check the compounds’ cytotoxic effect on normal and cancer cell lines. Finally, selected Ciprofloxacin derivatives were docked into the crystal structure of topoisomerase II (DNA gyrase) in complex with DNA (PDB ID: 5BTC).

We found a very interesting observation regarding results linked to Gram-positive stains. All monosubstituted compounds exhibited a broad and high spectrum of activity. Excluding derivative **2**, obtained minimal inhibitory concentrations (range 0.8–8 µg/mL) should be considered as very good. One of the compounds reached the antimicrobial potency level of reference Ciprofloxacin. Derivative **12** demonstrated activity against all standard Gram-positive *Staphylococci*, within the range of 0.8–1.6 µg/mL. As for Gram-positive hospital microorganisms, all tested derivatives were active. Measured MICs were in the range 1–16 µg/mL, confirming high antimicrobial potency. Compound **12** was recognized as a leading structure, with MICs 1 µg/mL for *S. pasteuri* KR 4358 and *S. aureus* T 5591. Disubstituted Ciprofloxacin derivatives **7**, **8**, **15**, **16** were rather inactive in this evaluation.

The conformational entropy of the docking results measured by the size of the largest cluster clearly favors the structures of the reference compound Ciprofloxacin, ligands with propyl (ligands **3**, **11**) and then a butyl linker (ligands **4**, **12**). Disubstituted derivatives were omitted during the early stage of the docking experiment. The preliminary analysis showed that substituents at the carboxyl residue (of the Ciprofloxacin scaffold) prevent the binding of disubstituted ligands to a tight pocket. More importantly, we need to emphasize that 3-oxo-4-carboxylic acid core is the active DNA-gyrase binding site and when structural changes were made in this fragment, there was an observed decrease in antibacterial potency.

Most of the tested derivatives exhibited moderate antiproliferative potency. However, three of the examined compounds (**3**, **11** and **16**) showed good activity against cancer cells, but were found not to be cytotoxic toward normal cells. Selectivity indexes were higher in every case comparing to reference. Doxorubicin SI were ranging 0.14–1.11 when above mentioned compounds 1.9–3.4.

Altogether, synthesized menthol and thymol Ciprofloxacin (N-4-piperazynyl) derivatives are promising antibacterials. Results showed high antimicrobial potency. From obtained group leading structures were established (**3**, **11**, **12** and **16**). All four will be transferred to more comprehensive evaluation toward wider panel of clinical strains, experiments related to damage of bacterial membrane proteins and in vitro study of inhibition of catalytic activities of bacterial topoisomerases. Further testing is needed for thesis conformation of dualistic mechanism of antimicrobial action.

## Data Availability

Not applicable.

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
