# Peer review of "Design and Synthesis of Menthol and Thymol Derived Ciprofloxacin: Influence of Structural Modifications on the Antibacterial Activity and Anticancer Properties"

_ijms, 2022, doi:10.3390/ijms23126600_

Round 1
Reviewer 1 Report
Review Report
The manuscript entitled ‘Design and synthesis of menthol and thymol derived ciprofloxacin. Influence of structural modifications on the antimicrobial activity’ authors have successfully synthesized a new series of Ciprofloxacin N-linked menthol and thymol hybrids and evaluated their antimicrobial and anticancer activity. Similar type of work has been reported by different researchers, where they modified the ciprofloxacin by incorporating the different pharmacophore at N or carboxylic acid end. The quality and quantum of the work is suitable for the journal but before acceptance authors must incorporate some comments given below.
1. Authors have not discussed the characterization of the new compounds in the results and discussion part, how they confirmed the formation of the compounds, need to discuss the characteristics signals in 1H and 13C NMR.
2. In the title, author mentioned antimicrobial activity but they evaluated the synthesized compounds only on bacterial strains. Antimicrobial is a vast topic and includes fungi also. Therefore I suggest the authors to mention antibacterial, not antimicrobial activity in the title. Also, they should include anticancer in the title.
3. In the abstract section, line 6, within the range of 0.8–1.6 g/mL and ----; authors should write 0.8–1.6 μg/mL.
4. Authors should write capital letter for all standard drugs mention in the manuscript.
5. Introduction section, page 1, last line, According to The ---------; authors should write According to the ----.
6. Page 2, According to The World Bank report ---; authors should write According to the World Bank report ---.
7. Page 3, Listeria [ 11-18]. The reference number needs to be corrected as [11-18]. Similarly reference [19-20], as [19-20].
8. Scheme 1, at page 5, need to be corrected, it is not clear, authors have used dichloromethane
And TEA, in the Scheme missing, authors should break the scheme into two parts for better understanding.
9. Page 6, the range of 0.8–1.6 g/mL.-----; authors should correct the range 0.8–1.6 μg/mL
10. In the 1H NMR experimental data of the compounds, authors should assign the signals with the no. of protons.
11. Authors should provide the NMR and Mass spectra of all the reported compounds as supplementary file.
Author Response
Thank you for your comments and help in improvment of our manuscript. Please see our responses below:
AD1. Suitable supplementary file was created with available data related to structural studies. Supplement contains part with intermediate substrates characterization and spectra for obtained derivatives.
AD2. Suggested change has been made. Anticancer properties of synthesized compounds will be continued and results disclosed in the separate paper.
AD3. Corrected.
AD4. Done.
AD5. Corrected.
AD6. Corrected.
AD7. Corrected.
AD8. Corrected.
AD9. Corrected.
AD9. Please see supplementary file.
AD10. Please see supplementary file.
Reviewer 2 Report
In this manuscript of "Design and synthesis of menthol and thymol derived ciprofloxacin. Influence of structural modifications on the antimicrobial activity", the authors successfully synthesized and characterized sixteen new ciprofloxacin menthol or thymol derivatives. All compounds were screened for antimicrobial activity and
in vitro cytotoxic properties against cancer cells and normal cells. Some compounds demonstrated activity against the standard bacteria strains, while three examined derivatives showed good activity against cancer cells but not cytotoxic toward normal cells.
However, the following concerns should be addressed for further publishing consideration.
1. In line 6 of abstract, the unit is ug/ml but not g/mL.
2. In section 3.2. Ciprofloxacin derivatives preparation, the amidation reaction is using triethylamine in CH2Cl2 under room temperature. However, the reaction condition in the Scheme 1 is not appropriate. Please optimize the Scheme 1 and illustrate the target compounds with exact menthol or thymol groups.
3. From 3.2. Ciprofloxacin derivatives preparation, please present the corresponding substrates, such as carboxylic acid chloride, appropriate menthol ester, and appropriate menthol ester, with structures or names. It will help the readers understand and repeat the work in the future.
4. Please provide citations about the control compound Doxorubicin.
5. Please apply the journal abbreviations for all the references.
Author Response
Thank you for your comments and help in improvment of our manuscript. Please see our responses below:
AD1. Corrected.
AD2. Suitable supplementary file was created with available data related to structural studies. Supplement contains part with intermediate substrates characterization and spectra for obtained derivatives.
AD3. Please see supplementary file.
AD4. Suitable citation was added.
AD5. Corrected.
Round 2
Reviewer 1 Report
The authors have incorporated the required suggestions and now is acceptable for publication in this journal.
Author Response
Thank you for your time and valuable comments.
Reviewer 2 Report
1. Your title only contained "Influence of structural modifications on the antibacterial activity". However, the new ligands' in vitro cytotoxic properties against cancer cells were an important section in this manuscript.
2. Please introduce the parent compound Ciprofloxacin as reference in Table 7 for comparison.
2. Some C-13 spectrum signals are too weak, such as compounds 7, 8, and 9. Please provide better ones.
Author Response
Thank you for all suggestions. We have improved our manuscript in line with your comments.
Title and Table 7 was corrected. The new spectra were recorded for compounds 7, 8 and 9 and based on this new data SI and experimental part of manuscript were updated. All suggested changes are in green in manuscript. Suitable new version of supplementary file was created, with new C-13 spectra.
Round 3
Reviewer 2 Report
Overall, the authors have done all the necessary correction and now the manuscript could be accepted in its current version.